# Exact Distributed Structure-Learning for Bayesian Networks

## Abstract

Learning the structure of a Bayesian network is currently practical for only a limited number of variables. Existing distributed learning approaches approximate the true structure. We present an exact distributed structure-learning algorithm to find a P-map for a set of random variables. First, by using conditional independence, the variables are divided into sets $\mathcal{X}_1, \ldots, \mathcal{X}_I$ such that for each $\mathcal{X}_i$, the presence and absence of edges that are adjacent with any interior node (a node that is not in any other $\mathcal{X}_j, j \neq i$) can be correctly identified by learning the structure of $\mathcal{X}_i$ separately without using the information of the variables other than $\mathcal{X}_i$. Second, constraint or score-based structure learners are employed to learn the P-map of $\mathcal{X}_i$, in a decentralized way. Finally, the separately learned structures are appended by checking a conditional independence test on the boundary nodes (those that are in at least two $\mathcal{X}_i$'s). The result is proven to be a P-map. This approach allows for a significant reduction in computation time, and opens the door for structure learning for a "giant" number of variables.

## 1 Introduction

Bayesian networks constitute a primary subfield within the realm of probabilistic graphical models, which serve as powerful tools for data modeling. These networks leverage directed acyclic graphs (DAGs) to represent probabilistic relationships in datasets. The process of structure learning in Bayesian networks involves the derivation of a DAG from empirical data (van den Boom et al., 2022). Two primary methodologies for learning the DAG from data are the constraint-based and score-based approaches (Kitson et al., 2021).

Constraint-based algorithms, such as PC algorithm (Spirtes et al., 2000), rely on the principles of sufficiency, Markov condition, and faithfulness assumption. These algorithms are designed to identify dependencies between variables without mediator variables. This is achieved by employing conditional independence (CI) tests (Guo et al., 2020). Score-based algorithms adopt an optimization-based strategy, wherein they define a likelihood function, often employing criteria like Bayesian Information Criterion (BIC). Both approaches yield a class of graphs known as independence-equivalent (I-equivalent) graphs, represented as partially Directed Acyclic Graphs (PDAGs) (Koller & Friedman, 2009).

Performing CI tests across all variables or optimizing the likelihood function over all potential graphs leads to computational challenges, often resulting in a computational explosion (Spirtes et al., 2000). This problem represents a significant challenge and limitation, particularly when dealing with a substantial number of variables (Peters et al., 2017; Ramsey et al., 2017). Several techniques have been developed to reduce the runtime, by for example, first running some fast conditional independence tests to quickly eliminate many edges in constraint-based algorithms (Giudice et al., 2022), limiting the conditioning set in the CI tests, (Sondhi & Shojaie, 2019), finding an order on the variables (Chen et al., 2019b;a) and (Gao et al., 2020), and parallelizing the CI tests (Zarebavani et al., 2019; Shahbazinia et al., 2023; Le et al., 2016).

Nevertheless, regardless of how much the speed of the structure learning algorithms are improved, their application will be limited to a small number of variables in practice. Score-based algorithms require an exhaustive search over the space of all DAGs, which is of size $\mathcal{O}(2^{n^2})$. Loading these many edges or DAGs on a single computing machine becomes readily infeasible for large values of $n$,

despite the many optimizations on reducing the order. As a result, existing computational resources are incapable to perform exact structure-learning on a "large" number of variables (Franzin et al., 2017), unless approximation techniques are used. Constraint-based algorithms, such as PC, start or interact with a fully connected network, which has $\mathcal{O}(n^2)$ edges for $n$ variables. This is more feasible to load on a single machine, however, the complexity of the algorithm itself is $\mathcal{O}(n^{p+2})$, where $p$ is the maximum number of parents of a variable in the "true" DAG (Koller & Friedman, 2009).

Reducing the structure-learning problem to several sub-problems that can be learned separately can be the key to solve this issue. An approximation distributed structure-learning approach was proposed in (Gu & Zhou, 2020), where the variables are partitioned into clusters that are learned in a distributed way and then appended to obtain the final DAG. Nevertheless, the result is an estimation of the true DAG and under the assumption of Gaussian-distributed variables.

The partitioning of variables is the main part of this approach. In many represented approaches, the resulting network by distributed learning is an approximation of a network that is obtained from centralized learning (Talvitie et al., 2019; Scanagatta et al., 2015). Additionally, in some other approaches (Xie et al., 2006) and (Liu et al., 2017), partitioning is performed using expert knowledge and requires conditional independence tests with high-order conditioning variables that cannot be used in many practical problems. (Zhang et al., 2020) proposed an optimization-based approach for partitioning using lower conditioning variables; however, the number of conditioning variables cannot be controlled.

We develop an exact distributed structure-learning algorithm that obtains the true P-map for a given set of random variables in three steps. First, the algorithm performs a *reduction* on the set of variables, by dividing them into sets $\mathcal{X}_1, \ldots, \mathcal{X}_I$. Each set $\mathcal{X}_i$ has a boundary $\mathrm{bd}(\mathcal{X}_i)$ that is the subset of nodes shared with other $\mathcal{X}_j$, i.e., $\cup_{j \neq i} \mathcal{X}_i \cap \mathcal{X}_j$, and an interior $\mathcal{X}_i^o$ which is the remainder, i.e., $\mathcal{X}_i \setminus \mathrm{bd}(\mathcal{X}_i)$. The reduction is such that the P-map confined to each set $\mathcal{X}_i$ is a *conditional P-map* for the marginal distribution of the variables in $\mathcal{X}_i$; namely, the presence and absence of all edges that are adjacent with the interior nodes of $\mathcal{X}_i$ are correctly learned by performing a structure-learner to find the P-map of $\mathcal{X}_i$. Roughly speaking, the "interior edges" of each set $\mathcal{X}_i$ can be learned separately, without the information about the nodes in the other $\mathcal{X}_j$'s. This naturally leads to the second step, where separate structure-learners, either constraint or score-based, are deployed to learn the local P-map structure of every $\mathcal{X}_i$. Finally, the local P-maps are concatenated to obtain the global P-map by performing a distributed PC-like algorithm on all boundary nodes. We prove that the resulting DAG is a P-map.

## 2 BACKGROUND

Consider a set of random variables $\mathcal{X} = \{X_1, \ldots, X_n\}$ with joint probability distribution $P$. Let $\mathcal{I}(P)$ denote the set of all conditional independencies implied by the distribution $P$, i.e., $\mathcal{I}(P) = \{(\mathcal{X}_1 \perp \mathcal{X}_2 \mid \mathcal{X}_3) : \mathcal{X}_1, \mathcal{X}_2, \mathcal{X}_3 \subseteq \mathcal{X}\}$. Let $\mathcal{G}$ be a DAG with node set $\mathcal{X}$. The DAG induces conditional independencies between the nodes using the notion of d-separation defined below. A *collider* in $\mathcal{G}$ is a triple of nodes $X_1 \to X_2 \leftarrow X_3$, where two of them are linked to the third. The collider is an *immorality* if the ending nodes $X_1$ and $X_3$ are not adjacent (connected). Three nodes are a *non-collider* if they do not form a collider.

**Definition 2.1** (d-separation). (Koller & Friedman, 2009) Consider the DAG $\mathcal{G}$ with node set $\mathcal{X}$. A trail (path) $\mathcal{T}$ between two nodes $X_1$ and $X_2$ in $\mathcal{X}$ is *active* relative to a set of nodes $\mathcal{Z}$ if *(i)* every non-collider on $\mathcal{T}$ is not a member of $\mathcal{Z}$, and *(ii)* every collider on $\mathcal{T}$ is an ancestor of some member of $\mathcal{Z}$. Otherwise, the trail is said to be *blocked* by $\mathcal{Z}$. The node subsets $\mathcal{X}_1$ and $\mathcal{X}_2$ are *d-separated* given the subset $\mathcal{Z}$, denoted $\mathrm{d\text{-}sep}_{\mathcal{G}}(\mathcal{X}_1, \mathcal{X}_2 \mid \mathcal{Z})$, if there is no active trail between any node $X_1 \in \mathcal{X}_1$ and any node $X_2 \in \mathcal{X}_2$ given $\mathcal{Z}$.

The set of all d-separations in $\mathcal{G}$ is denoted by $\mathcal{I}(\mathcal{G})$. We assume that for the distribution $P$, there exists a DAG $\mathcal{G}$ that satisfies both of the following well-known conditions: *(i)* Markovness, that is, $\mathcal{I}(\mathcal{G}) \subseteq \mathcal{I}(P)$, and *(ii* faithfulness, that is $\mathcal{I}(P) \subseteq \mathcal{I}(\mathcal{G})$. This results in $\mathcal{I}(P) = \mathcal{I}(\mathcal{G})$; namely, all conditional independencies in $P$ are captured by the d-separations in $\mathcal{G}$ and vice versa. DAG $\mathcal{G}$ is called a *P-map (perfect map)* for $P$. The problem is to find P-map $\mathcal{G}$ for distribution $P$. This problem is known as *structure learning*. There can be more than one P-map for a distribution $P$, e.g., two DAGs $\mathcal{G}_1$ and $\mathcal{G}_2$ where $\mathcal{I}(\mathcal{G}_1) = \mathcal{I}(\mathcal{G}_2) = \mathcal{I}(P)$. P-maps of the same distribution have the same skeleton and immoralities (Koller & Friedman, 2009). Consequently, the set of all P-maps for a distribution $P$ is represented by a *partially DAG (PDAG)* that is a graph over nodes $\mathcal{X}$ where two

nodes are adjacent, if they are adjacent in all of the P-maps and the connected edge is directed if all of the P-maps have the same direction, otherwise the edge is undirected. This PDAG is called the *P-map class PDAG for P*. The structure learning problem is often reduced to finding the P-map class PDAG for $P$.

*Problem* 1 (Structure learning). Consider the set of random variables $\mathcal{X}$ with distribution $P$ that admits a P-map. Find the P-map class PDAG for $P$.

Several *constraint-based algorithms*, such as Peter-Clark (PC) (Spirtes et al., 2000), and *score-based algorithms* with a *consistent score*, such as BIC, that perform an exhaustive search over the DAG space, are shown to solve Problem 1. We call an algorithm that solves Problem 1, a *P-map learner* (Koller & Friedman, 2009). Problem 1 is NP-hard and cannot be practically solved for a large number of variables $n$ (Koller & Friedman, 2009).

## 3 DISTRIBUTED STRUCTURE LEARNING

### 3.1 THE IDEA

Our goal is to solve Problem 1 in a distributed manner as explained intuitively below.

**Example 1.** *The DAG in figure 1 (a), denoted by $\mathcal{G}$, is a P-map for the joint distribution of random variables $X_1, \ldots, X_5$. Instead of learning the whole DAG at once, one can learn separately the P-map class PDAG of each of the sub-DAGs for $\mathcal{X}_1 = \{X_1, X_2, X_3\}$, $\mathcal{X}_2 = \{X_4, X_3\}$, and $\mathcal{X}_3 = \{X_5, X_3\}$ (figure 1 (b)), and then concatenate (and orient) them to obtain P-map $\mathcal{G}$ (figure 1 (c)). The reason is that each of the three subsets are d-separated, and hence, independent, from one another given their shared variable $X_3$, i.e.,*

$$X_1, X_2 \perp X_4 \mid X_3, \quad X_1, X_2 \perp X_5 \mid X_3, \quad X_4 \perp X_5 \mid X_3.$$

*Thus, when learning the structure of say the subset $\{X_1, X_2, X_3\}$, there is no active path between any of $X_1$ and $X_2$ to the other nodes (excluding $X_3$). This ensures two points. First, $X_1$ and $X_2$ are not connected by a path outside of $\{X_1, X_2, X_3\}$; that is, all of their dependencies are captured by this set. Hence, a structure learner can correctly learn the structure between these nodes without using the information from the other nodes $X_4$ and $X_5$. Second, when concatenating the graphs, no additional link between the subsets are needed. This idea does not apply to the partitioning subsets $\{X_1, X_4\}$ and $\{X_2, X_3, X_5\}$, because $X_1$ and $X_4$ do depend on $\{X_2, X_3, X_5\}$:*

$$\{X_1, X_4\} \not\perp \{X_2, X_3, X_5\}.$$

*Nodes $X_1$ and $X_4$ are related by a path outside of $\{X_1, X_4\}$, e.g., $X_1 \rightarrow X_3 \rightarrow X_4$. Hence, when learning the structure of $\{X_1, X_4\}$, the structure learner will incorrectly make $X_1$ and $X_4$ adjacent, because they are dependent. Similarly, subsets $\{X_1, X_3, X_4\}$ and $\{X_2, X_3, X_5\}$ would not work either. Because although the P-map of each subset can be learned correctly, the concatenation would require an additional link between $X_1$ and $X_2$ to recover $\mathcal{G}$.*

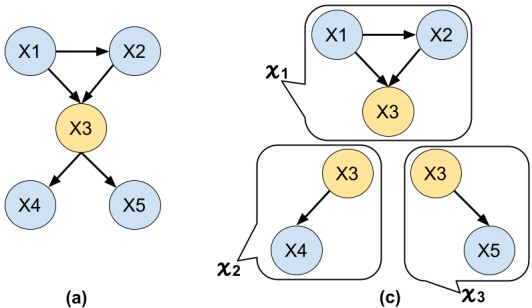

Figure 1: (a) A P-map for $\{X_1, \ldots, X_5\}$. (b) Three subsets that can be learned separately.

In what follows, we define mathematically the reduction approach taken in Example 1. For $\mathcal{Y} \subseteq \mathcal{X}$, let $P[\mathcal{Y}]$ denote the marginal probability distribution of variables $\mathcal{Y}$, and $\mathcal{G}[\mathcal{Y}]$ denote that sub-graph of

$\mathcal{G}$ limited to nodes $\mathcal{Y}$ and their connecting edges. The goal is to solve Problem 1 by an algorithm that is distributed over the nodes, that is, to divide nodes $\mathcal{X}$ into possibly overlapping subsets $\mathcal{X}_1, \ldots, \mathcal{X}_I$, so that every sub-graph $\mathcal{G}[\mathcal{X}_i]$, $i = 1, \ldots, I$, of the P-map class PDAG $\mathcal{G}$ for $P$ can be learned separately without using the information of the other nodes $\mathcal{X}_j, j \neq i$, and at the end to concatenate the sub-graphs so that the resulting is a P-map class PDAG for $P$.

The key step in this approach is the division of the nodes. Define a *cover* of $\mathcal{X}$ as a family of distinct nonempty subsets $\mathcal{X}_1, \ldots, \mathcal{X}_I \subseteq \mathcal{X}$ for some $I \geq 1$ such that $\cup_{i=1}^{I} \mathcal{X}_i = \mathcal{X}$. Define the *boundary of* $\mathcal{X}_i$, $i = 1, \ldots, I$, by $\mathrm{bd}(\mathcal{X}_i) = \mathcal{X}_i \cap (\cup_{j \neq i} \mathcal{X}_j)$, and the *interior of* $\mathcal{X}_i$ by $\mathcal{X}_i^o = \mathcal{X}_i \setminus \mathrm{bd}(\mathcal{X}_i)$. The union of all boundaries is called the *separator*, denoted $\mathcal{W} = \cup_i \mathrm{bd}(X_i)$. Correspondingly, the cover $\{\mathcal{X}_1, \ldots, \mathcal{X}_I\}$ for $\mathcal{X}$ is referred to as the *cover separated by* $\mathcal{W}$. In Example 1, $\mathcal{W} = \{X_3\}$. The union of arbitrary graphs $\mathcal{G}_1, \ldots, \mathcal{G}_I$, denoted $\cup_{i=1}^{I} \mathcal{G}_i$ is a graph with the node and edge set equal to the union of the nodes and edges of the graphs $\mathcal{G}_i$, and an edge $X - Y$ is directed from $X$ to $Y$ if it is so in every $\mathcal{G}_i$ that includes this edge; otherwise, it is undirected.

**Definition 3.1** (P-map reduction). Consider the set of random variables $\mathcal{X}$ with distribution $P$ that admits a P-map $\mathcal{G}$. Let $d \geq 1$ be an integer. A cover $\{\mathcal{X}_1, \ldots, \mathcal{X}_I\}$, $I > 1$, of $\mathcal{X}$ is a *(capped-d) P-map reduction* if for all $i = 1, \ldots, I$, (i) $|\mathcal{X}_i| \leq d$, (ii) $\mathcal{G}[\mathcal{X}_i]$ is a P-map for $P[\mathcal{X}_i]$, and (iii) for all $j \neq i$, there is no edge between $\mathcal{X}_i^o$ and $\mathcal{X}_j^o$ in $\mathcal{G}$.

Condition *(ii)* ensures that separate P-map learners can be used to learn the P-map class PDAG of each of the subsets $\mathcal{X}_i$. Namely, they can be learned in parallel and without communication, i.e., decentrally. Condition *(i)* restricts each subset $\mathcal{X}_i$ to include at most $d$ variables. The value of $d$ can be chosen based on the computational capacity of the P-map learners. Once all $\mathcal{G}[X_i]$'s are learned, Condition *(iii)* ensures that their union will be a P-map for the complete distribution $P$, and hence, solves Problem 1.

The cover in Example 1 is a P-map reduction (Remark A.2). Does a P-map reduction exist for every DAG? The answer is negative. For example in Figure 2-a, every pair of nodes are connected by two paths. Thus, to satisfy Condition *(iii)*, every element of a P-map reduction must be of size at least two. However, then at least one element of the cover violates Condition *(ii)*. For example, $\mathcal{G}[\mathcal{X}_1]$, which is the path $X_{12} \rightarrow X_{13} \leftarrow X_1 \rightarrow \ldots \rightarrow X_9$, is not a P-map for $P[\mathcal{X}_1]$ as that would require $X_9$ and $X_{12}$ to be adjacent.

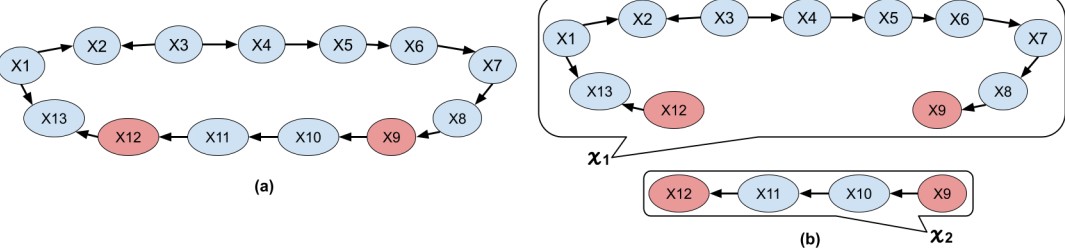

**(a)**

**(b)**

Figure 2: (a) The P-map $\mathcal{G}$ for variables $X_1, \ldots, X_{13}$. (b) The cover consisting of $\mathcal{X}_1 = \{X_9, X_{10}, X_{11}, X_{12}\}$ and $\mathcal{X}_2 = \{X_{12}, X_{13}, X_1, \ldots, X_9\}$, separated by $\mathcal{W} = \{X_9, X_{12}\}$.

Nevertheless, once $X_9$ and $X_{12}$ are observed, the path connecting any of the nodes $X_{13}, X_1, \ldots, X_8$ to either of $X_{10}$ and $X_{11}$ is blocked. Namely, the interior of the cover element $\mathcal{X}_1$ becomes d-separated given its boundary $X_9$ and $X_{12}$. Consequently, every d-separation in the sub-DAG confined to $\mathcal{X}_1$, i.e., $\mathcal{G}[\mathcal{X}_1]$, either itself exists in $\mathcal{I}(P)$ or when it is additionally conditioned to the boundary variables $\mathrm{bd}(\mathcal{X}_1)$. On the other hand, the partitioning of $\mathcal{X}$ into the cover elements does not cause the loss of a d-separation in the resulting sub-DAGs, i.e., they all remain faithful. This motivates the following definitions.

**Definition 3.2** (Conditional P-map). Let $\mathcal{X}$ be a set of random variables with distribution $P$ and consider subset $\mathcal{Z} \subseteq \mathcal{X}$. DAG $\mathcal{G}$ defined over $\mathcal{X}$ is a *conditional P-map for $P$ given $\mathcal{Z}$* if (i) $P$ is faithful to $\mathcal{G}$, and (ii) $\mathcal{G}$ is a *conditional I-map* for $P$; that is, if d-sep$_{\mathcal{G}}(\mathcal{X}_1, \mathcal{X}_2 \mid \mathcal{X}_3)$, $\mathcal{X}_1, \mathcal{X}_2, \mathcal{X}_3 \subseteq \mathcal{X}$, then there exists $\mathcal{Z}_0 \subseteq \mathcal{Z}$ such that $\mathcal{X}_1 \perp \mathcal{X}_2 \mid \mathcal{X}_3 \cup \mathcal{Z}_0$.

**Definition 3.3** (Conditional P-map reduction). Consider the set of random variables $\mathcal{X}$ with distribution $P$ that admits a P-map $\mathcal{G}$. Let $d \geq 1$ be an integer. A cover $\{\mathcal{X}_1, \ldots, \mathcal{X}_I\}$, $I > 1$, of $\mathcal{X}$ is a *(capped-$d$) conditional P-map reduction* if for all $i = 1, \ldots, I$, *(i)* $|\mathcal{X}_i| \leq d$, *(ii)* $\mathcal{G}[\mathcal{X}_i]$ is a conditional P-map for $P[\mathcal{X}_i]$ given $\mathrm{bd}(\mathcal{X}_i)$, and *(iii)* for all $j \neq i$, there is no edge between $\mathcal{X}_i^o$ and $\mathcal{X}_j^o$ in $\mathcal{G}$.

According to Condition *(ii)* in Definition 3.3, every conditional independence in $\mathcal{I}(P[\mathcal{X}_i])$ is included in $\mathcal{I}(\mathcal{G}[\mathcal{X}_i])$. This ensures that constraint-based algorithms, such as PC, will not incorrectly eliminate an edge when learning the structure of $\mathcal{X}_i$. On the other hand, every d-separation in $\mathcal{I}(\mathcal{G}[\mathcal{X}_i])$ exists in $\mathcal{I}(P[\mathcal{X}_i])$ either itself or when some of the boundary nodes $\mathrm{bd}(\mathcal{X}_i)$ are additionally observed (see Remark A.3 in the Appendix for why the second case does not always hold). This ensures that PC can correctly identify the edges that do not exist between two interior nodes or an interior node and a boundary node in $\mathcal{G}[\mathcal{X}_i^o]$. Therefore, the P-map structure of the interiors $\mathcal{X}_i^o$ and their connections to the boundary nodes $\mathrm{bd}(\mathcal{X}_i)$ can be learned in a decentralized way. Although the intra and inter connections of the boundaries $\mathrm{bd}(\mathcal{X}_i)$ cannot be learned decentrally and generally requires information from the all of the elements. Moreover, Condition *(iii)* ensures that no appending between the interiors is required to obtain the P-map for $\mathcal{X}$. In the following subsection, we explain how to learn the conditional P-maps and the structure of the boundaries of conditional P-map cover, yet we will first focus on finding the cover.

*Problem* 2. Given integer $d \geq 0$ and set of random variables $\mathcal{X}$ with distribution $P$ that admits a P-map, find a capped-$d$ conditional P-map reduction for $\mathcal{X}$.

The idea in Example 1 and Figure 2 to solve Problem 2 was to divide the nodes into subsets that are d-separated given their common nodes. More specifically, we need a separator $\mathcal{W}$ and a partition of the set $\mathcal{X} \setminus \mathcal{W}$ into some subsets $\mathcal{C}_1, \ldots, \mathcal{C}_I$ that are pairwise independent conditioned on $\mathcal{W}$.

**Definition 3.4** (Separated-by cover). Consider random variables $\mathcal{X}$ with distribution $P$ and a subset $\mathcal{W} \subset \mathcal{X}$. The *cover for $\mathcal{X}$ separated by $\mathcal{W}$* is a collection of sets $\{\mathcal{W} \cup \mathcal{C}_i\}_{i=1}^I$, $I \geq 1$, such that *(i)* $\{\mathcal{C}_1, \ldots, \mathcal{C}_I, \mathcal{W}\}$ is a partition for $\mathcal{X}$, *(ii)* $\mathcal{C}_i \perp \mathcal{C}_j \mid \mathcal{W}$ for all distinct $i, j = 1, \ldots, I$, *(iii)* and $I$ is maximal.

To solve Problem 2, one can iteratively apply separators to the elements of a cover until they no longer decompose. How to find the cover separated by $\mathcal{W}$? Consider an order for variables $\mathcal{X}$, represented by vector $\boldsymbol{X} = [X_1, \ldots, X_n]^\top$. Vector $\boldsymbol{X}_{\mathcal{W}}$ is defined as $\boldsymbol{X}$ where the elements of $\mathcal{W}$ are removed and let $[\boldsymbol{X}_{\mathcal{W}}]_i$ be the $i^{\mathrm{th}}$ entry of $\boldsymbol{X}_{\mathcal{W}}$. For subset $\mathcal{W} \subset \mathcal{X}$, define the symmetric $(n - |\mathcal{W}|) \times (n - |\mathcal{W}|)$ *dependency matrix* $D_{\mathcal{W}}$ by $D_{\mathcal{W}}(i, j) = 1$ if $[\boldsymbol{X}_{\mathcal{W}}]_i \not\perp [\boldsymbol{X}_{\mathcal{W}}]_j \mid \mathcal{W}$ and otherwise $D_{\mathcal{W}}(i, j) = 0$ for all $i, j \in \{1, \ldots, n - |\mathcal{W}|\}$, with $D_{\mathcal{W}}(i, i) = 1$ for all $i$. By using a permutation matrix $P$, we have $\bar{D}_{\mathcal{W}} = P D_{\mathcal{W}} P^{-1}$ where $\bar{D}_{\mathcal{W}}$ is the block diagonal form of $D_{\mathcal{W}}$. Then each group of entries of the transformed vector $\bar{\boldsymbol{X}}_{\mathcal{W}} = P \boldsymbol{X}_{\mathcal{W}}$ that correspond to a block of $\bar{D}_{\mathcal{W}}$ constitutes one of the desired partitions, which combined with $\mathcal{W}$ form an element of the P-map reduction (see Example 1-revisited in the appendix). Nevertheless, finding the permutation matrix can be computationally costly. An alternative is to treat the dependency matrix as an adjacency matrix, defining an undirected graph and find the connected components of this graph. This can be done in $\mathcal{O}(n^2)$ (Cormen et al., 2001).

## 3.2 THE ALGORITHMS

We provide Algorithm 1 as the distributed learning algorithm to solve Problem 1. The algorithm consists of three sub-algorithms: One that performs a conditional P-map reduction (solves Problem 2), one that learns each of the elements of the reduction (interiors and between interiors and boundaries), and finally, one to append the elements of the reduction (learning the boundaries).

---

**Algorithm 1:** Distributed structure learner

---

**Input:** Set of random variables $\mathcal{X}$ with joint probability distribution $P$ that admit a P-map; the maximum number of variables in each element of the cover, $d$; the maximum number of separator variables, $W$

**Output:** A P-map $\mathcal{G}$ for P

1  $\{\mathcal{X}_1, \ldots, \mathcal{X}_I\} \leftarrow$ Algorithm 2 $(d, W)$;
                              `// Alternatively, Algorithm 3 can be used.`

2  **for** $i = 1, \cdots, I$
3    |   $\mathcal{G}_i \leftarrow$ P-map learner$(\mathcal{X}_i)$
4  $\bar{\mathcal{G}} = \bigcup\limits_{i=1}^{I} \mathcal{G}_i$
5  **for** $i = 1, \cdots, I$
6    |   $\mathcal{G} \leftarrow$ Boundary PC $(\mathcal{X}_i, \bar{\mathcal{G}})$

---

We provide Algorithms 2 and 3 for the first part, i.e., to solve Problem 2. Both are based on the idea to iteratively find separators that would decompose the components of a conditional P-map reduction of $\mathcal{X}$ into another reduction.

Algorithm 2 goes through the cover $\mathcal{X}_\mathcal{I}$ (initially set to $\{\mathcal{X}\}$), picks the greatest component $\mathcal{U} \in \mathcal{X}_\mathcal{I}$, and checks if any subset $\mathcal{W} \subset \mathcal{U}$ separates the component into a cover of cardinality greater than one (which is a conditional P-map reduction for $\mathcal{U}$). If so, then the algorithm updates the cover $\mathcal{X}_\mathcal{I}$ by replacing $\mathcal{U}$ with its cover and moves to the next greatest component in $\mathcal{X}_\mathcal{I}$. This process continues until either all components of the cover have a size less than $d$ or none of the components can be further reduced. The notation $\mathcal{P}(\mathcal{U})$ is the power set of the set $\mathcal{U}$, i.e., the set of all subsets of $\mathcal{U}$.

---

**Algorithm 2:** Parallel conditional P-map reduction finder CI based

---

**Input:** $\mathcal{X} = \{X_1, \cdots, X_n\}$, $d$, $W$, and the number of processors $N_p$.

**Output:** A conditional P-map reduction $\mathcal{X}_\mathcal{I}$ of $\mathcal{X}$

1  $\mathcal{X}_\mathcal{I} \leftarrow \{\mathcal{X}\}$;
2  $w \leftarrow 0$;                                `// w:the size of the separator`
3  $\mathcal{K} \leftarrow \emptyset$
4  **while** $\max_{\mathcal{U} \in \mathcal{X}_\mathcal{I}} |\mathcal{U}| > d$ *and* $w \leq W$ **do**
5    |   $\mathcal{X}'_\mathcal{I} \leftarrow \mathcal{X}_\mathcal{I}$;      `// X'_I: potentially decomposable cover members (w.r.t. w)`
6    |   **while** $\mathcal{X}'_\mathcal{I} \neq \emptyset$ *and* $\max_{\mathcal{U}' \in \mathcal{X}'_\mathcal{I}} |\mathcal{U}'| > d$ **do**
7    |  |   $\mathcal{U} \leftarrow \arg\max_{\mathcal{U}' \in \mathcal{X}'_\mathcal{I}} |\mathcal{U}'|$;          `// U: greatest cover member`
8    |  |   **for** $\mathcal{M} \subset \mathcal{P}(\mathcal{U}) \setminus \mathcal{K}$ *and* $|\mathcal{M}| = N_p$ *and* $|\mathcal{W}| = w$ *for* $\mathcal{W} \in \mathcal{M}$
9    |  |  |   **for** $\mathcal{W} \in \mathcal{M}$
10   |  |  |  |   $\{\mathcal{W} \cup \mathcal{C}_\mathcal{W}^i\}_{i=1}^{I_\mathcal{W}} \leftarrow$ the cover for $\mathcal{U}$ separated by $\mathcal{W}$;
11   |  |  |   $\mathcal{K} \leftarrow \mathcal{K} \cup \mathcal{M}$
12   |  |  |   **if** $\sum\limits_{\mathcal{W} \in \mathcal{M}} I_\mathcal{W} > N_p$           `// the cover was a reduction`
13   |  |  |  |   $\mathcal{X}_\mathcal{I} \leftarrow$ Cover finding for $\mathcal{U}$ by intersection method on $\{\mathcal{W} \cup \mathcal{C}_\mathcal{W}^i\}_{i=1}^{I_\mathcal{W}}$ if $I_\mathcal{W} > 1$;
                         `// update the cover`
14   |  |  |  |   $\mathcal{X}_\mathcal{I} \leftarrow \mathcal{X}_\mathcal{I} \setminus \{\mathcal{U}\}$
15   |  |  |  |   $\mathcal{X}'_\mathcal{I} \leftarrow \mathcal{X}_\mathcal{I}$
16   |  |  |  |   Break;
17   |  |   $\mathcal{X}'_\mathcal{I} \leftarrow \mathcal{X}'_\mathcal{I} \setminus \{\mathcal{U}\}$
18   |   $w \leftarrow w + 1$;

---

The other conditional P-map finder is Algorithm 3. The problem with Algorithm 2 is that it may run too many CI tests. When searching for separators of size $w$ of a cover element $\mathcal{U}$ with cardinality $u$, all $\binom{u}{w}$ subsets of $\mathcal{U}$ are checked for being a separator; next all $\binom{u}{w+1}$ subsets are checked and so on. However, in Algorithm 3, once a separator of size $w$ is found, the algorithm searches for all single nodes to be added to this separator, those are, $\binom{u-w}{1}$; next all $\binom{u-w}{2}$, and so on, until another reduction happens. The problem with Algorithm 3 is that the number of conditioning variables in the

---

**Algorithm 3:** Parallel Non-monotone conditional P-map reduction finder CI based

**Input:** $\mathcal{X} = \{X_1, \cdots, X_n\}$, $d$, $W$, and the number of processors $N_p$.
**Output:** A conditional P-map reduction $\mathcal{X}_\mathcal{I}$ of $\mathcal{X}$

1   $\mathcal{X}_\mathcal{I} \leftarrow \{\mathcal{X}\}$;
2   $w \leftarrow 0$;                     `// w:the size of the separator`
3   **while** $w \leq W$ *and* $\max_{\mathcal{U} \in \mathcal{X}_\mathcal{I}} |\mathcal{U}| > d$ **do**
4      $\mathcal{X}'_\mathcal{I} \leftarrow \mathcal{X}_\mathcal{I}$;        `// `$\mathcal{X}'_\mathcal{I}$`: potentially decomposable cover members`
      `(w.r.t. w)`
5      $\mathcal{K} \leftarrow \emptyset$;
6      **while** $\mathcal{X}'_\mathcal{I} \neq \emptyset$ *and* $\max_{\mathcal{U}' \in \mathcal{X}'_\mathcal{I}} |\mathcal{U}'| > d$ **do**
7          $\mathcal{U} = \arg\max_{\mathcal{U}' \in \mathcal{X}'_\mathcal{I}} |\mathcal{U}'|$;           `// `$\mathcal{U}$`: greatest cover member`
8          $\mathrm{bd}(\mathcal{U}) \leftarrow \bigcup_{\mathcal{U}' \in \mathcal{X}_\mathcal{I} \setminus \{\mathcal{U}\}} \mathcal{U} \cap \mathcal{U}'$
9          **for** $\mathcal{M} \subset \mathcal{P}(\mathcal{U} \setminus (\mathrm{bd}(\mathcal{U}) \cup \mathcal{K}))$ *and* $|\mathcal{M}| = N_p$ *and* $|\mathcal{W}| = w$ *for* $\mathcal{W} \in \mathcal{M}$
10             **for** $\mathcal{W} \in \mathcal{M}$
11                $\mathcal{X}_\mathcal{W} \leftarrow \{\mathcal{U}\}$
12                $\{\mathcal{W} \cup \mathrm{bd}(\mathcal{U}) \cup \mathcal{C}^i_{\mathcal{W} \cup \mathrm{bd}(\mathcal{U})}\}^{I_\mathcal{W}}_{i=1} \leftarrow$ the cover for $\mathcal{U}$ separated by $\mathcal{W} \cup \mathrm{bd}(\mathcal{U})$;
13                **if** $I_\mathcal{W} > 1$                 `// the cover was a reduction`
14                    **for** $U \in \mathrm{bd}(\mathcal{U})$
15                       **for** $i = 1, \cdots, I_\mathcal{W}$
16                          **if** $U \not\perp \mathcal{C}^i_{\mathcal{W} \cup \mathrm{bd}(\mathcal{U})} | \mathcal{W} \cup \mathrm{bd}(\mathcal{U}) \setminus \{U\}$
17                            $\mathcal{C}^i_\mathcal{W} \leftarrow \{U\} \cup \mathcal{C}^i_{\mathcal{W} \cup \mathrm{bd}(\mathcal{U})}$
18                    **if** $\mathrm{bd}(\mathcal{U}) \setminus \cup \mathcal{C}^i_\mathcal{W} \neq \emptyset$
19                      $\mathcal{X}_\mathcal{W} \leftarrow \{\mathcal{W} \cup (\mathrm{bd}(\mathcal{U}) \setminus \cup \mathcal{C}^i_\mathcal{W})\}$;
20                  $\mathcal{X}_\mathcal{W} \leftarrow (\mathcal{X}_\mathcal{W} \setminus \{\mathcal{U}\}) \cup \{\mathcal{W} \cup \mathcal{C}^i_\mathcal{W}\}^{I_\mathcal{W}}_{i=1}$;
21             **if** $\sum_{\mathcal{W} \in \mathcal{M}} I_\mathcal{W} > N_p$            `// the cover was a reduction`
22                $\mathcal{X}_\mathcal{I} \leftarrow$ Covering $\mathcal{U}$ by intersection method on $\mathcal{X}_\mathcal{W}$ if $I_\mathcal{W} > 1$;    `// update`
                `the cover`
23                $\mathcal{X}_\mathcal{I} \leftarrow \mathcal{X}_\mathcal{I} \setminus \{\mathcal{U}\}$
24                $\mathcal{X}'_\mathcal{I} \leftarrow \mathcal{X}_\mathcal{I}$
25                $w \leftarrow 1$
26                Break;
27             **else**
28                $\mathcal{K} \leftarrow \mathcal{K} \cup \mathcal{M}$
29          $\mathcal{X}'_\mathcal{I} \leftarrow \mathcal{X}'_\mathcal{I} \setminus \{\mathcal{U}\}$
30      $w \leftarrow w + 1$;

---

CI tests grows quickly, because the separators are never removed from the conditioning part. Namely, once a cover element is reduced into another cover by a separator of size $w$, the next reduction will require a CI test with a conditioning of size at least $w + 1$ as both the previous separator and a new separator of size one will be conditioned on. This does not happen in Algorithm 2; namely, once a separator of size $w$ is found, the CI test required for finding the next separator will have again $w$ conditioning variables, as the algorithm does not condition on the previously found separator. For the same reason, Algorithm 3 may be unable to find a capped-$d$ cover for small values of $d$.

After the cover finding process, a structural learning method is individually applied to each set of covering variables, resulting in the discovery of the network structure for each subset of variables. Let $\{\mathcal{X}_1, \cdots, \mathcal{X}_I\}$ denote a covering of $\mathcal{X}$ that is derived from the algorithms 2 and 3. Each subset of variables, $\mathcal{X}_i$ for $i = 1, \cdots, I$, can be independently learned using either score-based or constraint-based algorithms. This leads to the identification of the local structures, denoted as $\mathcal{G}_i$ for $i = 1, \cdots, I$. In $\mathcal{G}_i$, the all edges between every two interior variables, and between a variable in the interior and another in the boundary set equal to the edges of a P-map $\mathcal{G}$ for $P$. In the process of constructing the comprehensive network structure, the local structures obtained in the prior stage must be concatenated together. Each cover set shares a common variable set, denoted as $\mathrm{bd}(\mathcal{X}_i)$, representing the observed

variables that correspond to specific nodes in the local networks. By learning the edges between the boundary variables and assembling the local networks using these common nodes, the comprehensive network structure for all variables is constructed.

---

**Algorithm 4:** The Boundary PC Algorithm

---

**Input:** The union of $\{\mathcal{G}_i\}_{i=1}^I$, cover sets $\{\mathcal{X}_i\}_{i=1}^I$ and joint probability distribution $P$
**Output:** A P-map $\bar{\mathcal{G}}$ for $P$

1  $\mathrm{Sep}(X, Y) = \emptyset$ for all $X, Y \in \mathrm{bd}(\mathcal{X}_i)$;
2  **for** $X \in \mathrm{bd}(\mathcal{X}_i)$
3      **for** $Y \in \mathrm{bd}(\mathcal{X}_i) \cap \mathrm{Adj}(\bar{\mathcal{G}}, X)$
4          $\mathcal{U} = (\bigcup_{X \in \mathcal{X}_k, Y \notin \mathcal{X}_k} \mathrm{bd}(\mathcal{X}_k)) \cup (\bigcup_{Y \in \mathcal{X}_k, X \notin \mathcal{X}_k} \mathrm{bd}(\mathcal{X}_k)) \setminus \{X, Y\}$
5          **if** $X \perp Y \mid \mathcal{U}$
6              Remove the edge $X - Y$ from $\bar{\mathcal{G}}$;
7              $\mathrm{Sep}(X, Y) \leftarrow \mathcal{U}$;
8  For $X - Z - Y$ that $X, Y \in \mathrm{bd}(\mathcal{X}_i)$ are not adjacent in $\bar{\mathcal{G}}$, if $Z \notin \mathrm{Sep}(X, Y)$ then orient $X - Z - Y$ as an immolarity $X \to Z \leftarrow Y$.
9  Orient the other edges using the orientation rules in (Spirtes et al., 2000).

---

### 3.3 THE SUPPORTING THEORY

**Proposition 3.5.** *Consider random variables $\mathcal{X}$ with distribution $P$. A cover $\mathcal{X}_\mathcal{I} = \{\mathcal{X}_1, \ldots, \mathcal{X}_I\}$ satisfying*

$$\forall i, j \neq i \qquad \mathcal{X}_i \perp \mathcal{X}_j \mid \mathrm{bd}(\mathcal{X}_i) \tag{1}$$

*is a conditional P-map reduction.*

**Proposition 3.6.** *Consider random variables $\mathcal{X}$ with distribution $P$ that admits a P-map class PDAG $\mathcal{G}$. Assume that $\mathcal{X}_\mathcal{I} = \{\mathcal{X}_i\}_{i=1}^I$ is a conditional P-map reduction for $\mathcal{X}$, and consider an arbitrary $i \in \{1, \ldots, I\}$. Two nodes $X_1, X_2 \in \mathcal{X}_i$ where at least one of which is from $\mathcal{X}_i^o$ are adjacent (resp. non-adjacent) in $\mathcal{G}[\mathcal{X}_i]$ if and only if they are adjacent (resp. non-adjacent) in the P-map class PDAG of $P[\mathcal{X}_i]$. Moreover, every triple of nodes $X_1, X_2, Z \in \mathcal{X}_i$, where at least two of which are in $\mathcal{X}_i^o$, form an immolarity in $\mathcal{G}$ if and only if they do so in the P-map class PDAG of $P[\mathcal{X}_i]$. Finally, if a triple of nodes $X_1, X_2, Z \in \mathcal{X}_i$, where $X_1 \in \mathcal{X}_i^o$, form an immolarity $X_1 \to Z \leftarrow X_2$ in the P-map class PDAG of $P[\mathcal{X}_i]$, then the immolarity also exists in $\mathcal{G}$.*

**Lemma 3.7.** *The cover output by Algorithms 2 and 3 satisfies Condition equation 1.*

**Lemma 3.8.** *Consider random variables $\mathcal{X}$ with distribution $P$ that admits a P-map class PDAG $\mathcal{G}$. Assume that $\mathcal{X}_\mathcal{I} = \{\mathcal{X}_i\}_{i=1}^I$ is a conditional P-map reduction for $\mathcal{X}$, and consider an arbitrary $J \subseteq \{1, \ldots, I\}$. Two nodes $X_1, X_2 \in \mathrm{bd}(\mathcal{X}_j)$ for all $j \in J$, are non-adjacent in $\mathcal{G}$ if and only if they are non-adjacent in a P-map class PDAG of $P[\mathcal{X}_{j_0}]$ for at least a $j_0 \in J$.*

**Lemma 3.9.** *Under Algorithms 2 and 3, every edge in $\mathcal{G}_i$ belongs to a cover component.*

**Theorem 3.10.** *Algorithm 1 outputs a P-map for $P$.*

*Proof.* It suffices to prove that the output of the algorithm, say $\hat{\mathcal{G}}$, has the same skeleton and immolarities as the P-map PDAG class for $P$, say $\mathcal{G}$. In view of Lemma 3.7, Proposition 3.5, Algorithm 1 outputs a cover $\mathcal{X}_\mathcal{I} = \{\mathcal{X}_i\}_{i=1}^I$ that is a conditional P-map reduction. Hence, the interior nodes of no two elements of the cover are adjacent in both $\hat{\mathcal{G}}$ and $\mathcal{G}$. On the other hand, Proposition 3.6 guarantees that all edges between every interior node of element $\mathcal{X}_i$ and another node in $\mathcal{X}_i$ are correctly identified for every $i$. So it only remains to show that the algorithm also correctly identifies the edges between the boundary nodes of every $\mathcal{X}_i$. Denote by $\mathcal{G}'$ the graph obtained by Algorithm 1, before executing (sub-)Algorithm 4. Assume that there is an edge between two boundary nodes $X, Y \in \mathcal{X}_i$ in $\mathcal{G}'$ that does not exist in $\mathcal{G}$. Since the edge does not exist in the P-map $\mathcal{G}$, there is some $\mathcal{U} \in \mathcal{X}$ such that $X_1 \perp X_2 \mid \mathcal{U}$. In view of Lemma A.4, the set $\mathcal{U}$ can be chosen such that all nodes in $\mathcal{U}$ are either adjacent with $X_1$ or adjacent with $X_2$ in $\mathcal{G}$. Denote the nodes adjacent with $X_1$ (resp. $X_2$) in $\mathcal{G}$ by $\mathcal{N}_{X_1}^{\mathcal{G}}$ (resp. $\mathcal{N}_{X_2}^{\mathcal{G}}$). It follows from Lemma 3.9 that $\mathcal{N}_{X_i}^{\mathcal{G}} = \cup_{j=1}^I \mathcal{N}_{X_i}^{\mathcal{G}[\mathcal{X}_j]}$ for $i = 1, 2$. Since $\mathcal{G}'$ includes all of the edges in $\mathcal{G}$, it follows that the the set $\mathcal{U}$ can be found by searching through

the union of the neighbors of $X_1$ (resp. $X_2$) in each $\mathcal{X}_j$, which is what Algorithm 4 does. Hence, the edge will be detected and eliminated from $\mathcal{G}'$. Finally, Algorithm 1 does not delete an edge that actually exists in $\mathcal{G}$ as the elimination of an edge in Algorithm 4 happens only if a conditional independence holds between the variables.

Table 1: The results for cover finding algorithms. The notation $I$ is the number of cover elements and $\ell_{max}$ is the cardinality of the greatest cover element.

| DATASET | # NODES | $d$ | # CPUs | $\ell_{max}$ | $I$ | # RUNTIME(ALG. 1) | # RUNTIME (PC) |
|---------|---------|-----|--------|--------------|-----|-------------------|----------------|
| ASIA | 8 | 6 | 20 | 6 | 3 | 0.87 | 1.16 |
| SACHS | 11 | 8 | 30 | 4 | 7 | 28.6 | 50 |
| CHILD | 20 | 15 | 30 | 14 | 7 | 14 | 127.2 |
| INSURANCE | 27 | 20 | 30 | 25 | 3 | 86.5 | 208 |
| WATER | 32 | 24 | 30 | 26 | 7 | 15.1 | 25.2 |
| MILDEW | 35 | 26 | 30 | 34 | 2 | 532 | 858 |
| ALARM | 37 | 27 | 30 | 30 | 5 | 48 | 84.4 |
| BARLEY | 48 | 36 | 30 | 47 | 2 | 929 | 1293 |
| HAILFINDER | 56 | 42 | 30 | 56 | 1 | 72257 | 90821 |
| HEPAR2 | 70 | 52 | 30 | 51 | 17 | 632 | 1496 |
| WIN95PTS | 76 | 57 | 30 | 65 | 8 | 398 | 358 |

As with the immoralities, it follows from Proposition 3.6, that every immorality in $\mathcal{G}'$ with at least one interior node of some cover element $\mathcal{X}_i$ also exists in $\mathcal{G}$. Now if the edges of an immorality in $\mathcal{G}'$ with all three nodes being a boundary node of some cover element $\mathcal{X}_i$ are not eliminated in $\hat{\mathcal{G}}$, then the immorality also exists in $\mathcal{G}$. So all immoralities in $\mathcal{G}'$ that also appear in $\hat{\mathcal{G}}$ belong to $\mathcal{G}$. On the other hand, if an immorality emerges after executing Algorithm 4, i.e., it belongs to $\hat{\mathcal{G}}$ but not $\mathcal{G}'$, then it should also belong to $\mathcal{G}$, because Algorithm 4 is basically the PC algorithm that starts from the graph $\mathcal{G}'$ that is a superset $\mathcal{G}$ (and PC is known to correctly identify the immoralities). Therefore, every immorality in $\hat{\mathcal{G}}$ is included in $\mathcal{G}$. Now we show that that every immorality in $\mathcal{G}$ is included in $\hat{\mathcal{G}}$. In view of Lemma 3.9 every edge is on a cover element. Moreover, it is impossible to have three boundary nodes $X$, $Y$, and $Z$ forming a collider $X \to Z \leftarrow Y$, and the three nodes do not belong to the same cover element, because then the element including $X$ and that including $Y$ will not be d-separated conditioned on $Z$. Hence, according to Proposition 3.6, we only need to show that the immoralities in $\mathcal{G}$ with all three nodes belonging to the boundary of some element, or when exactly one node is an interior and the other two are the boundary of the same element. The proof of the first part is similar to the previous case (Algorithm 4 being basically the PC algorithm) and it checks the existence of every boundary edge. For the second part, we have a node $X \in \mathcal{X}_i^o$ for some $i$, and two boundary nodes $Y, Z \in \text{bd}(\mathcal{X}_i)$, such that $Y \to X \leftarrow Z$ is an immorality in $\mathcal{G}$. If the immorality also exists in $\mathcal{G}'$, there is nothing to prove. Otherwise, $Y$ and $Z$ are adjacent in $\mathcal{G}'$, but the edge will be eliminated by Algorithm 4 and then checked for such immorality. This completes the proof. $\square$

## 4 EXPERIMENTS

We compared the performance of Algorithm 1 and PC on the datasets ASIA (Lauritzen & Spiegelhalter, 1988), ALARM (Beinlich et al., 1989), INSURANCE (Binder et al., 1997), CHILD (Spiegelhalter & Cowell, 1992), WATER (Jensen et al., 1989), HAILFINDER (Abramson et al., 1996), HEPAR2 (Andreassen et al., 1989). The number of samples for all datasets is 10,000. The computations were performed on a system with 2 xAMD Rome 7532@ 2.4GHz 256M cache. Algorithm 3 was employed as a sub-algorithm within Algorithm 1, while the PC algorithm was used for the local structure learners. The value of $d$ was set to 0.75 times the number of variables, and $W$ was set to 1. The runtime for both Algorithm 1 and the PC algorithm is reported in Table 1. The number of CPUs was set to 30 for all datasets, except for ASIA, where fewer CPUs were used due to the low number of variables. According to the Wilcoxon signed-rank test, Algorithm 1 was significantly faster, up to 2 times ($p$-value = 0.01) compared to the PC algorithm. Additionally, as shown in Table 2, the structural Hamming distance indicates that the error is not significantly different between Algorithm 1 and the PC algorithm.

Table 2: Structural Hamming Distance

| DATASET | ALG. 1 | PC |
|---|---|---|
| ASIA | 0 | 0 |
| SACHS | 0 | 0 |
| CHILD | **0** | 1 |
| INSURANCE | 15 | **14** |
| WATER | 37 | **36** |
| MILDEW | **10** | 11 |
| ALARM | 3 | 3 |
| BARLEY | 28 | 28 |
| HAILFINDER | 52 | 52 |
| HEPAR2 | **57** | 64 |
| WIN95PTS | 42 | **41** |

## 5 CONCLUSION

We developed a distributed approach for structure learning applicable to both constraint-based and score-based algorithms. The main concept is to identify a cover set for the set of variables using conditional independence (CI) tests. Two key parameters, the upper bound of the cardinality of cover elements $d$ and the number of conditioning variables $W$ play crucial roles in determining an appropriate cover set. Reducing the value of $d$ while increasing $W$ can decrease the cardinality of the greatest cover element and increase the number of cover elements. However, this adjustment may lead to an increase in the number of CI tests, which in turn could raise the runtime of the cover-finding algorithms. Algorithms 2 and 3 are executed in parallel across multiple CPUs. Consequently, increasing the number of CPUs can help reduce runtime. Thus, it is essential to select the values of $d$, $W$, and the number of CPUs carefully to ensure that the cardinality of the greatest cover element remains small, enabling the runtime to be less than that of the standard version of the structure learning algorithm.

One might argue that increasing the number of CPUs, only to reduce runtime by a factor of two, might seem like an inefficient use of resources. However, it is important to recognize that in the realm of parallel computation, particularly across nodes, alternatives for comparison are limited. Existing methods either parallelize the CI tests for each edge, which still requires substantial memory to load the entire graph, or they depend on expert knowledge to inform the process. Our approach complements these by focusing on breaking the graph into manageable pieces, allowing any of these methods to be applied efficiently to the cover elements. Furthermore, this process can now occur in parallel across CPUs, which are generally more cost-effective and accessible than GPUs. This flexibility not only broadens the applicability of our approach but also makes it feasible in a wider range of computational environments.

The proposed approach results in exact distributed structure learning algorithms. Specifically, it has been demonstrated that the output of Algorithm 1 yields the exact structure without any approximation in cover finding, local structure learning, and the concatenation of local structures. In addition, unlike other exact distributed algorithms (Xie et al., 2006) and (Liu et al., 2017), which rely on expert knowledge and conditional independence tests with high-order conditioning variables, the proposed approach utilizes only a low-order conditioning set bounded by $W$.

In summary, our distributed structure learning approach efficiently handles large Bayesian networks by breaking the problem into smaller, manageable components, allowing for parallel execution on multiple CPUs. This method achieves exact results without requiring expert knowledge or high-order conditioning, offering a practical solution to the scalability issues in traditional algorithms. By reducing memory demands and enabling flexible integration with existing techniques, our approach enhances computational efficiency while preserving accuracy, making it a valuable tool for large-scale structure learning across diverse domains. As computational demands continue to grow, this work lays a strong foundation for the scalable and accurate learning of complex probabilistic models.

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

## A  APPENDIX

The following definition is a reformulation of Definition 2 in terms of P-map class PDAGs.

**Definition A.1.** Consider integer $d \geq 0$ and the set of random variables $\mathcal{X}$ with distribution $P$ that admits a P-map, and let $\mathcal{G}$ be the P-map class PDAG for $P$. A cover $\{\mathcal{X}_1, \ldots, \mathcal{X}_I\}$ of $\mathcal{X}$ is a $d$-capped P-map reduction if for every $i = 1, \ldots, I$, *(i)* $|\mathcal{X}_i| \leq d$ and *(ii)* the P-map class PDAG for $P[\mathcal{X}_i]$ equals $\mathcal{G}[\mathcal{X}_i]$, and *(iii)* the P-map class PDAG for $P$ equals $\cup_{i=1}^{I} \mathcal{G}[\mathcal{X}_i]$.

*Remark* A.2. Let $\mathcal{I}(\mathcal{G})[\mathcal{Y}]$ denote those conditional independencies in $\mathcal{I}(\mathcal{G})$ that are over nodes $\mathcal{Y}$, i.e., $\mathcal{I}(\mathcal{G})[\mathcal{Y}] = \{(\mathcal{X}_1 \perp \mathcal{X}_2 \mid \mathcal{X}_3) \in \mathcal{I}(\mathcal{G}) : \mathcal{X}_1, \mathcal{X}_2, \mathcal{X}_3 \subseteq \mathcal{Y}\}$. In Example 1, the cover $\{\{X_1, X_2, X_3\}, \{X_4, X_3\}, \{X_5, \mathcal{X}_3\}\}$ (figure 1 (b)) is a capped-$d$ P-map reduction for $\{X_1, \ldots, X_5\}$ for any $d \geq 3$. Conditions *(i)* and *(iii)* in Definition 3.1 are clearly met. For Condition *(ii)*, we show that the set of conditional independencies of each subset in the cover, e.g., $\{X_1, X_2, X_3\}$, matches the set of d-separations of the corresponding sub-graph, i.e., $\mathcal{I}(P[X_1, X_2, X_3]) = \mathcal{I}(\mathcal{G}[X_1, X_2, X_3])$. According to the d-separations in $\mathcal{G}$, $\mathcal{I}(\mathcal{G}[X_1, X_2, X_3]) = \mathcal{I}(\mathcal{G}[X_4, X_3]) = \mathcal{I}(\mathcal{G}[X_5, X_3]) = \emptyset$. On the other hand, since $\mathcal{G}$ is a P-map for $P$, $\mathcal{I}(P[X_1, X_2, X_3]) = \mathcal{I}(\mathcal{G})[X_1, X_2, X_3] = \emptyset$, $\mathcal{I}(P[X_4, X_3]) = \mathcal{I}(\mathcal{G})[X_4, X_3] = \emptyset$, and $\mathcal{I}(P[X_5, X_3]) = \mathcal{I}(\mathcal{G})[X_5, X_3] = \emptyset$.

*Remark* A.3. In Figure A.3, every pair of $\mathcal{X}_i$'s are d-separated given the union of the boundary nodes $\mathcal{W}$, so the $\mathcal{X}_i$'s can be shown to be a conditional P-map reduction. Now the interior nodes $X_1$ and $X_3$ in $\mathcal{X}_2$ are not d-separated given the boundary nodes $\{X_2, X_5, X_{12}\}$. However, $X_1$ and $X_3$ are d-separated in the sub-graph $\mathcal{G}[\mathcal{X}_2]$. This is why enforcing the boundary nodes to be always observed does not help to find the d-separations of the sub-graphs of the cover elements $\mathcal{G}[\mathcal{X}_i]$ – it may be that only the d-separation itself appears in $\mathcal{I}(P)$.

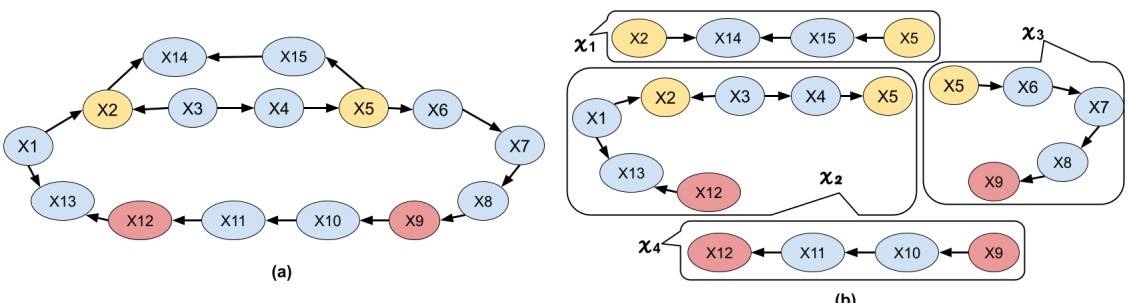

Figure 3: (a) The P-map $\mathcal{G}$ for variables $X_1, \ldots, X_{15}$. (b) The cover consisting of $\mathcal{X}_1 = \{X_5, X_{15}, X_{14}, X_2\}$, $\mathcal{X}_2 = \{X_{12}, X_{13}, X_1, \ldots, X_5\}$, $\mathcal{X}_3 = \{X_5, \ldots, X_9\}$, and $\mathcal{X}_4 = \{X_9, X_{10}, X_{11}, X_{12}\}$.

**Example 1** (revisited). *Let* $\mathcal{W} = \{X_3\}$ *and consider the vector* $\boldsymbol{X} = [X_1, X_2, X_3, X_4, X_5]^\top$. *Then* $\boldsymbol{X}_{\{X_3\}} = [X_1, X_2, X_4, X_5]^\top$. *The dependency matrix then equals*

$$\bar{D}_{\{X_3\}} = D_{\{X_3\}} = \begin{bmatrix} 1 & 1 & 0 & 0 \\ 1 & 1 & 0 & 0 \\ 0 & 0 & 1 & 0 \\ 0 & 0 & 0 & 1 \end{bmatrix}$$

*which is already in a block-diagonal form. Hence, the P-map reduction consists of* $\{X_1, X_2\} \cup \{X_3\}$, $\{X_4\} \cup \{X_3\}$, *and* $\{X_5\} \cup \{X_3\}$. *Should, instead, the order* $\boldsymbol{X} = [X_1, X_4, X_2, X_3, X_5]^\top$ *was used, yielding* $\boldsymbol{X}_{\{X_3\}} = [X_1, X_4, X_2, X_5]^\top$, *then* $\bar{D}_{\{X_3\}}$ *would be obtained as above by using the following permutation matrix applied to the dependency matrix* $D_{\{X_3\}}$:

$$D_{\{X_3\}} = \begin{bmatrix} 1 & 0 & 1 & 0 \\ 0 & 1 & 0 & 0 \\ 1 & 0 & 1 & 0 \\ 0 & 0 & 0 & 1 \end{bmatrix}, \quad P = \begin{bmatrix} 1 & 0 & 0 & 0 \\ 0 & 0 & 1 & 0 \\ 0 & 1 & 0 & 0 \\ 0 & 0 & 0 & 1 \end{bmatrix}.$$

---

**Algorithm 5:** The PC Algorithm

---

**Input:** A Covering set $\mathcal{X}_i$ and their joint probability distribution $P$
**Output:** An undirected graph

1 Form the complete undirected graph $\mathcal{G}_i$ over nodes $\mathcal{X}_i$;
2 $\mathrm{Sep}(X, Y) = \emptyset$ for all $X, Y \in \mathcal{X}_i$;
3 $m = 0$;
4 **while** *maximum node degree in $\mathcal{G}_i$ is greater than $m$* **do**
5     **for** $X \in \mathcal{X}_i$
6         **for** $Y \in \mathrm{Adj}(\mathcal{G}_i, X)$
7             **for** $\mathcal{U} \subseteq \mathrm{Adj}(\mathcal{G}_i, X) \setminus \{Y\}$ *and* $|\mathcal{U}| = m$
8                 **if** $X \perp Y \mid \mathcal{U}$
9                     Remove the edge $X - Y$ from $\mathcal{G}_i$;
10                     $\mathrm{Sep}(X, Y) \leftarrow \mathcal{U}$;
11     $m = m + 1$;
12 Orient the edges using the orientation rules in (Spirtes et al., 2000).

---

**Lemma A.4.** *(Based on (Pearl, 2009)) Consider random variables $\mathcal{X}$ with joint distribution $P$ that admits a P-map $\mathcal{G}$. Vertices $X$ and $Y$ are not adjacent in $\mathcal{G}$ if and only if $X \perp Y \mid \mathcal{U}$ for $\mathcal{U} = \mathrm{Pa}_X$ (parents of $X$ in $\mathcal{G}$) or $\mathrm{Pa}_Y$ (parents of $Y$ in $\mathcal{G}$).*

**Lemma A.5.** *(Based on (Koller & Friedman, 2009)) Let $\mathcal{G}$ be a P-map of a distribution $P$ and assume that $X, Y$ and $Z$ are a potential immorality, i.e., $X$ and $Y$ are not adjacent but both are adjacent with $Z$. Then $X, Y, Z$ form an immorality, i.e., $X \rightarrow Z \leftarrow Y$ if and only if $X \not\perp Y \mid \mathcal{U}$ for any set $\mathcal{U} \ni Z$.*

**Lemma A.6.** *(Based on (Koller & Friedman, 2009)) Let $\mathcal{G}$ be a P-map of a distribution P, and assume that there exists three nodes $X, Y, Z$, where $X$ and $Y$ are adjacent with $Z$ but with themselves, and the three do not form an immorality, i.e., $X \rightarrow Z \leftarrow Y$ is not in $\mathcal{G}$. If $\mathcal{U}$ is such that $X \perp Y \mid \mathcal{U}$, then $Z \in \mathcal{U}$.*

**Proof of Proposition 3.5** Let $\mathcal{W} = \mathrm{bd}(\mathcal{X}_i)$. Condition *(iii)* in Definition 3.3 follows the fact that $\mathcal{X}_i^o \perp \mathcal{X}_j^o \mid \mathcal{W}$ and the fact that two nodes are not adjacent in a P-map should they be conditionally independent. So it suffices to prove Condition *(ii)*. It is straightforward to show that $\mathcal{G}[\mathcal{X}_i]$ is faithful to $P[\mathcal{X}_i]$ for every $i = 1, \ldots, I$: the subgraph $\mathcal{G}[\mathcal{X}_i]$ is obtained by removing some nodes and edges from the P-map $\mathcal{G}$, which does not add a new path between two nodes; so nodes without a connecting path in $\mathcal{G}$ remains so in $\mathcal{G}[\mathcal{X}_i]$. Now we show that $\mathcal{G}[\mathcal{X}_i]$ is a conditional I-map for $P$. Consider the d-separation d-$\mathrm{sep}_{\mathcal{G}[\mathcal{X}_i]}(\mathcal{Y}_1, \mathcal{Y}_2 \mid \mathcal{Y}_3)$, where $\mathcal{Y}_1, \mathcal{Y}_2, \mathcal{Y}_3 \subseteq \mathcal{X}_i$. Let $\mathcal{W}_v \subseteq \mathcal{W}$ denote the set of separator nodes that form a collider with a node in $\mathcal{Y}_1$ and a node in $\mathcal{Y}_2$ or are a descendent node of such a collider. Define $\mathcal{W}_n = \mathcal{W} \setminus \mathcal{W}_v$. We prove by contradiction that d-$\mathrm{sep}_{\mathcal{G}}(\mathcal{Y}_1, \mathcal{Y}_2 \mid \mathcal{Y}_3 \cup \mathcal{W}_n)$, where $\mathcal{Y}_1, \mathcal{Y}_2, \mathcal{Y}_3 \subseteq \mathcal{X}_i$. Assume the contrary, implying that there is an active path $\mathcal{T}$ from a node $Y_1 \in \mathcal{Y}_1$ to a node $Y_2 \in \mathcal{Y}_2$ when observing $\mathcal{W}_n$. Path $\mathcal{T}$ cannot include any of the nodes $\mathcal{W}_n$ as they would block the path. Also, since observing $\mathcal{W}_n$ does not activate any collider, $\mathcal{T}$ must include a node $S \notin \mathcal{X}_i$ out of $\mathcal{X}_i$. On the other hand, the separator drives the cover elements independent, yielding $\mathcal{X}_i \perp S \mid \mathcal{W}$, meaning that the nodes in $\mathcal{W}$ block all paths such as $\mathcal{T}$ that leave $\mathcal{X}_i$ and have two end nodes in $\mathcal{X}_i$. Since $\mathcal{T}$ does not include $\mathcal{W}_n$, it includes some of the nodes in $\mathcal{W}_v$. Namely, path $\mathcal{T}$ leaves $\mathcal{X}_i$ from a node $W_1 \in \mathcal{W}_v$, reaches $S$ and returns to $\mathcal{X}_i$ by another node $W_2 \in \mathcal{W}_v$. Hence, for $\mathcal{T}$ to be active, its edge adjacent to $W_1$ must be an outgoing edge and the same holds for $W_2$. However, then that part of path $\mathcal{T}$ with ends $W_1$ and $W_2$ that passes through $S$ will have a collider which blocks the whole path $\mathcal{T}$, a contradiction, completing the proof. $\square$

**Proof of Proposition 3.6** In view of Lemma A.4, if $X_1$ and $X_2$ are adjacent in $\mathcal{G}[\mathcal{X}_i]$, then they are not independent conditioned on any subset of other values, including those in $\mathcal{X}_i$. Hence, the dependence also reveals in $P[\mathcal{X}_i]$, implying the existence of the link in the P-map class PDAG of $P[\mathcal{X}_i]$. If $X_1$ and $X_2$ are not adjacent in $\mathcal{G}[\mathcal{X}_i]$, then they are independent conditioned on some subset $\mathcal{U} \in \mathcal{X}$. On the other hand, $\mathcal{X}_i^o \perp (\mathcal{X} \setminus \mathcal{X}_i) \mid \mathrm{bd}(\mathcal{X}_i)$, implying that $\mathrm{bd}(\mathcal{X}_i)$ blocks all paths between $\mathcal{X}_i^o$ and nodes other than $\mathcal{X}_i$. On the other hand, similar to the proof of Proposition 3.5 it can be shown that the above independence also holds when we only condition on those boundary nodes $\mathcal{W}_n$ that do not form a collider with $X_1$ and $X_2$ and are not a descendent node that would activate such a collider, i.e., $\mathcal{X}_i^o \perp (\mathcal{X} \setminus \mathcal{X}_i) \mid \mathcal{W}_n$. Thus, $X_1 \perp X_2 \mid \mathcal{U}$ yields $X_1 \perp X_2 \mid (\mathcal{X}_i \cap \mathcal{U}) \cup \mathcal{W}_n$ as those nodes of $\mathcal{U}$ that are out of $\mathcal{X}_i$ and have an active path to $X_1$ or $X_2$, their path can be blocked by

observing $\mathcal{W}_n$. Hence, there $X_1$ and $X_2$ become independent also by conditioning on nodes that are only in $\mathcal{X}_i$. Therefore, in view of Lemma A.4, they will not be adjacent in the P-map class PDAG of $P[\mathcal{X}_i]$.

Now we prove the second part. Suppose that $X_1$ and $X_2$ form an immorality with another node $Z \in \mathcal{X}_i$ in $\mathcal{G}[\mathcal{X}_i]$, i.e., $X_1 \rightarrow Z \leftarrow X_2$, and that at least two of $X_1$ , $X_2$, and $Z$ are in $\mathcal{X}_i^o$. Then $X_1$ and $X_2$ are not d-separated in $\mathcal{G}[\mathcal{X}_i]$ given any $\mathcal{U} \subseteq \mathcal{X}_i$ that contains $Z$. This implies that $X_1$ and $X_2$ are not d-separated in $\mathcal{G}$ given any $\mathcal{U} \subseteq \mathcal{X}$ that contains $Z$ as adding more edges and vertices to $\mathcal{G}[\mathcal{X}_i]$ does not make an already active path inactive. Now due to $\mathcal{G}$ being a P-map class PDAG for $P$, it holds that $X_1 \not\perp X_2 \mid \mathcal{U}$ for any $\mathcal{U} \subseteq \mathcal{X}$ that contains $Z$. On the other hand, based on what we proved earlier, $X_1$ and $X_2$ are connected to $Z$ and are not adjacent with each other in the P-map PDAG class of $P[\mathcal{X}_i]$. Hence, in view of Lemma A.5, $X_1$ and $X_2$ form an immorality with $Z$ in the P-map. Now suppose that $X_1$ and $X_2$ are not adjacent, both connected to $Z \in \mathcal{X}_i$, do not form an immorality in $\mathcal{G}[\mathcal{X}_i]$, and that again at least two of $X_1$ , $X_2$, and $Z$ are in $\mathcal{X}_i^o$. Clearly, the same holds in $\mathcal{G}$. In view of Lemma A.6, if $X_1 \perp X_2 \mid \mathcal{U}$ for some $\mathcal{U} \subseteq \mathcal{X}$, then $Z \in \mathcal{U}$. Thus, if $X_1 \perp X_2 \mid \mathcal{U}$ for some $\mathcal{U} \subseteq \mathcal{X}_i$, then $Z \in \mathcal{U}$, meaning that the condition holds also in $P[\mathcal{X}_i]$, which completes the proof according to Lemma A.5.

Now consider the triple $X_1$, $X_2$, and $Z$, where only one of them, say $X_1$, is in $\mathcal{X}_i^o$ and the other two are in $\mathrm{bd}(\mathcal{X}^i)$. Consider the case where the three nodes form the immorality $X_1 \rightarrow Z \leftarrow X_2$ in the P-map PDAG class of $P[\mathcal{X}_i]$. Then there exists a $\mathcal{U} \subseteq \mathcal{X}_i$ not including $Z$, such that $X_1 \perp X_2 \mid \mathcal{U}$, which implies that there is no active path between $X_1$ and $X_2$ that has a node out of $\mathcal{X}_i$. We prove by contradiction that $X_2$ and $Z$ are adjacent in $\mathcal{G}$. Otherwise, there exists an active path $\mathcal{T}$ of length at least two between $X_2$ and $Z$ regardless of whether any subset $\mathcal{U} \subseteq \mathcal{X}_i$ is observed. Therefore, every node in $\mathcal{T}$ is out of $\mathcal{X}_i$. Let $V \in \mathcal{T}$ be the node in $\mathcal{T}$ that is adjacent to $Z$. The direction of the edge between $V$ and $Z$ cannot be from $Z$ to $V$, because then by observing both $Z$ and the aforementioned $\mathcal{U}$, $X_1$ and $X_2$ will become d-separated, which is impossible. For the same reason, $X_1$ is linked to $Z$. Hence, $X_1$, $Z$, and $V$ form the collider $X_1 \rightarrow Z \leftarrow V$, implying that $X_1 \not\perp V \mid Z$. This, however, contradicts equation 1. Hence, $X_2$ and $Z$ are adjacent in $\mathcal{G}$. Then the immorality $X_1 \rightarrow Z \leftarrow X_2$ exists in $\mathcal{G}$ as well as otherwise, there cannot exist a $\mathcal{U} \subseteq \mathcal{X}_i$ not including $Z$ such that $X_1 \perp X_2 \mid \mathcal{U}$, a contradiction. $\square$

**Proof of Lemma 3.7** We prove by induction on the cardinality $k$ of the cover, where $k = K_1, K_2 \dots$. For both algorithms, the base case $k = K_1 > 1$ holds trivially. Assume that the result holds for $k = m$. Consider that iteration in the algorithms where the cover has cardinality $m$, denoted by $\{\mathcal{X}_1, \dots, \mathcal{X}_m\}$ and let element $\mathcal{X}_i$ be the next cover that will be reduced. According to equation 1, $\mathcal{X}_i \perp \mathcal{X}_j \mid \mathrm{bd}(\mathcal{X}_i)$. This implies that the boundary nodes of $\mathcal{X}_i$, block every path that connect the interior nodes of $\mathcal{X}_i$ to other elements of the cover. In Algorithm 2, $\mathcal{X}_i$ will be reduced to a cover $\{\mathcal{W} \cup \mathcal{C}_{\mathcal{W}}^i\}_{i=1}^I$ where $\mathcal{C}_{\mathcal{W}}^i \perp \mathcal{C}_{\mathcal{W}}^j \mid \mathcal{W}$ for all $i \neq j$. Now consider an arbitrary $\mathcal{C}_i$. Should $\mathrm{bd}(\mathcal{X}_i) \subseteq \mathcal{W}$, then $\mathcal{C}_{\mathcal{W}}^i \perp \mathcal{X}_j | \mathcal{W}$ for all $j$. Otherwise, some of the nodes in $\mathrm{bd}(\mathcal{X}_i)$ are in $\cup_{j \neq i} \mathcal{C}_{\mathcal{W}}^j$, and hence, are d-separated from $\mathcal{C}_{\mathcal{W}}^i$ after observing $\mathcal{W}$. In other words, $\mathcal{W}$ either directly or indirectly blocks all the paths from $\mathcal{C}_{\mathcal{W}}^i$ to $\mathcal{X}_j$ for every $j \neq i$. This is because observing $\mathcal{W}$ does not activated any collider that would in turn activate a path between $\mathcal{X}_i$ and $\mathcal{X}_j$ (every node in $\mathcal{X}_i$ that is adjacent to another $\mathcal{X}_j$ is included in $\mathrm{bd}(\mathcal{X}_i)$ as otherwise equation 1 is violated). This completes the proof for Algorithm 2. The proof for Algorithm 3 is similar. $\square$