# OpenReview forum: "Exact Distributed Structure-Learning for Bayesian Networks"
_ICLR.cc/2025/Conference — Submitted to ICLR 2025_

### Official Review · Reviewer_skzp · 2024-10-27

**Soundness:** 2
**Presentation:** 3
**Contribution:** 3
**Rating:** 5
**Confidence:** 4

**Summary:**

This paper proposes an exact distributed structure learning method. The proposed method first divides the variables into a set of groups of variables by using conditional independence tests in order to make sure that the edges connecting interior variables in each group can be solely identified without accessing the variables in other groups. The authors provide theoretical proof of their theory and evaluate their method on some datasets. The results show that the proposed algorithm outperforms the classic PC algorithm in some of the tested datasets.

**Strengths:**

The work is well written, and the motivation is clear and sufficient. The paper proposes a novel exact structure learning algorithm with solid theoretical guarantees. Compared with other exact learning algorithms, the proposed method requires less computational time.

**Weaknesses:**

1. Although the authors claim in the abstract that "This approach allows for a significant reduction in computation time, and opens the door for structure learning for a “giant” number of variables.". No BN with a "giant" number of variables was tested in the experiment. It would be more convincing if the author could evaluate the proposed method in larger BNs.
2. There is a lack of comparison with other score-based algorithms, especially other exact score-based algorithms that also guarantee the return of the true PDAG, such as GOBNILP.
3. The experimental results in Table 2 show that there is merely a slight difference between the performance of ALG. 1 and PC, which is insufficient to prove the effectiveness of the proposed algorithm.

**Questions:**

1. In the second example (Figure 2 on page 4), if we separate the variables as {X1, X2, X3}, {X1, X13, X2}, and a series of consecutive two nodes groups, for example {X3, X4}, {X4, X5}, {X5, X6}, etc. Isn't this separation a valid P-map reduction? Does it violate any of the three criteria in Definition 3.1?
2. Does the conditional P-map reduction exist for every DAG?

---

### Official Review · Reviewer_uhxa · 2024-10-29

**Soundness:** 2
**Presentation:** 1
**Contribution:** 2
**Rating:** 3
**Confidence:** 4

**Summary:**

The paper tackles the problem of structure learning of Bayesian networks with many variables. Structure learning is currently only done for a limited number of variables and is usually an approximation of the true graph. The authors present an exact distributed structure-learning algorithm to solve this by splitting the problem into several smaller parts as follows. The variables are divided into overlapping subsets (this division is called a "cover"). The division is done by using conditional independence such that edges can be correctly identified by learning a graph on each subset separately. The separated parts are joined using further conditional independence tests. Empirically, the proposed algorithm is found to improve the PC algorithm.

**Strengths:**

The presentation in sections 1 and 2 is nicely to-the-point. The examples in section 3 help to get an intuitive feeling. Distributed algorithms for structure learning are going to be an important step in tackling the computational hardness of the problem; compared to existing work, the exactness of the algorithm proposed here is a significant advantage.

**Weaknesses:**

**Problematic explanation** Section 3 was overall very unclear, to such an extent that I can't evaluate its correctness. It starts (in section 3.1) with an intuitive sketch of how the algorithms will work, and already provides some formal definitions, before the algorithms are given in section 3.2 and the proof of correctness in section 3.3. However, the intuitions and definitions given in section 3.1 do not align with how things actually work in the later sections. In particular, definition 3.1 (P-map reduction) gives conditions that a cover can satisfy; according to the paragraph below the definition, for such covers we can learn a graph on each subset, take their union (as defined just above definition 3.1), and this union will be the graph we are looking for (a P-map). This is incorrect, as demonstrated by these two counterexamples:

- The distribution has P-map A -> B <- C, and the cover is {{A,B}, {B,C}}. This is a P-map reduction, but the union will not have directed arrows.
- The distribution has P-map with the three arrows A->B, A->C and A->D, and the cover is {{A,B},{A,C},{D}}. Again this is a P-map, but the union won't have a edge between A and D.

It might be the case that the actual proof of correctness later in the paper is not affected by these counterexamples (though I note that both counterexamples also meet definition 3.3: Conditional P-map reduction). But even if that is the case, the explanation in section 3.1 is not helpful but actually harms the reader's understanding of why the algorithms work and what the role of these definitions is. The actual conditions necessary on the cover aren't specified anywhere that I could see, so that I don't know how to review the correctness of Algorithm 4. (There are also problems with the clarify of the algorithms themselves; I list several questions about these below.)

**Experiments** While the algorithm can be combined with any structure-learning algorithm, the empirical results only use the classical PC algorithm. The PC algorithm is also the only baseline algorithm that is compared against. A convincing empirical evaluation would consider a wider range of algorithms beyond just this one. In particular, an algorithm should be included that does not make the restrictive faithfulness assumption. (Contrary to what is stated on line 035, more recent algorithms use weaker assumptions instead of faithfulness.) Another limitation is that the experiments use $W=1$. (I didn't see this explained in the text, but from reading Algorithm 2, $W$ is apparently the maximum number of nodes in the separator.) So only the most simple setting of the algorithm seems to have been evaluated.

**Relation to tree decomposition** The work seems related to treewidth-based structure learning algorithms, but no such methods are referenced. See for instance the relevant section in the survey by Scanagatta, Salmerón and Stella (2019). The decompositions considered in the paper under review seem to be of a very specific form: all components must consist of a subset of nodes unique to that component, and a subset that is shared by *all* components (Definition 3.4). The notion of the tree decomposition that underlies the definition of treewidth provides a substantial generalization of what decompositions are allowed. In view of this, I find this paper's concept of decomposition limits its significance.

**Questions:**

- Regarding related work, (Xie et al., 2006) and (Liu et al., 2017) are competing approaches for which it is mentioned that they require impractically large conditioning sets, and something similar is said about (Zhang et al., 2020). Can you demonstrate that your algorithm improves over those in this regard?

- Experimental results: What is the motivation for choosing these values of $W$ and $d$? Can you say why the Structural Hamming Distance of HEPAR2 is 7 points lower for Algorithm 1 than for PC? And why the running time on WIN95PTS is higher?

- In algorithms 2 and 3: What is $N_p$? What is the "intersection method"? What exactly is line 8 in algorithm 2 iterating over?

- In algorithm 4:

  - in line 1, the $i$ isn't quantified over. Should this say that this line iterates over all $X,Y$ that occur *together* in some cover set's boundary?

  - in line 2, what is the role of $i$? Should this be: for $(X,i)$ s.t. $X \in bd(\mathcal{X}_i)$?

  - in line 4, I think you mean that both unions range over all $k$ such that ..., correct? (If so, write "$k:$" at the start of the subscript.)

**Other remarks:**

- "Concatenate" and "append" are used in many places when talking about graphs. These are well-defined as operations on sequences, but it is unclear what they mean when applied to graphs.

- The paper frequently uses the word "adjacent" to describe a relation between nodes and edges; this relation is called "incidence"/"incident".

- "without mediator variables" (line 036) is probably not what you meant to say.

- Score-based approached do not require an exhaustive (i.e. brute-force) search (line 053)!

- Figure 1 has no subfigure (b), only (a) and (c). (Yet (b) is referenced somewhere.)

- An underspecified definition of "cover separated by W" is given before definition 3.1, but this term isn't used until after the actual definition (3.4) is given.

- Below definition 3.4, I find the story about the permutation matrix not very helpful. I'd recommend removing it and going straight to the explanation in terms of connected components.

- Lemma 3.8 doesn't seem to be proved nor used anywhere.

- Please don't position Table 1 in the middle of the proof.

- Reference Chen et al. (2019) is duplicated.

- line 650: I think Definition 2 should be Problem 2.

---

### Official Review · Reviewer_nUZE · 2024-10-30

**Soundness:** 3
**Presentation:** 3
**Contribution:** 3
**Rating:** 8
**Confidence:** 2

**Summary:**

This paper studies structure learning for faithful DAG models under distributed learning setup. It proposes an distributed algorithm to recover the P-map of the underlying distribution exactly, which has three steps: decompose the whole set of variables into subsets of variables (covers) via so-called conditional P-map reduction; perform structure learning for each of the cover in a parallel manner; concatenate the learned P-maps for the covers together and further figure out the edges between the nodes on the boundary of the covers. The proposed algorithm is shown to recover the P-map consistently. Experiemnts are conducted compared with classic PC algorithm.

**Strengths:**

- The paper is well written with illustrating examples, helpful pause and summary of goals.
- Modern structure learning is restricted when huge number of nodes is present. This paper explores distributed learning as a way out, and rigorously provides an approach to effectively conduct distributed learning for structure learning by decomposing the problem into more tractable subproblems.
- Experiments on several datasets show competitive accuracy compared to PC algorithm, and faster runtime, demonstrating the promising performance of distributed learning.

**Weaknesses:**

- Algorithms 2 and 3 are dense in notations, which are hard to parse and understand the intuition. While several sentences are available to describe the algorithms, more details are desired to clearly convey the message. Maybe a running example is helpful to understand the algorithms.
- The theoretical results in Section 3.3 are stated without further explanation. The detailed proof can be moved to appendix, and replaced by the implication of each lemma and a proof outline of the Theorem 3.10.
- More benchmarks should be added in the experiments, at least GES.

**Questions:**

- The step to find the separating covers seems to be time consuming. In Algorithm 2, at the beginning when $\mathcal{U}=\mathcal{X}$,  we need to consider the power set of all variables as candidates, which is also exponential in number of total variables, which is comparable to PC algorithm. How do we understand the tradeoff?
- Is there any time complexity result? e.g. dependence on number of variables, maximum degree, input parameters $d,W$.
- The improvement in runtime in Table 1 is not significant compared to PC. They seem to be on the same order. Any explanation on that?
- What is $\mathcal{X}_k$ in Algorithm 4?

---

### Official Review · Reviewer_Lzgk · 2024-11-03

**Soundness:** 3
**Presentation:** 3
**Contribution:** 2
**Rating:** 5
**Confidence:** 4

**Summary:**

The authors study the Bayesian network learning problem and propose an algorithm that first divides the original problem into several smaller subproblems, solves each subproblem using either a score- or constraint-based method, and then combines the solved subproblems to form the final p-map. They provide theoretical guarantees and empirical evaluations.

**Strengths:**

This problem is relevant to the causal inference literature.
 The paper is well-written, with necessary concepts clearly explained.
The proposed method is particularly interesting as it breaks the problem into smaller subproblems, potentially leading to more efficient learning algorithms for large networks.
The theoretical results are valuable and could contribute to future developments.

**Weaknesses:**

Although the proposed learning algorithm is novel and theoretically sound, its efficiency and scalability are not analyzed. The first part of the proposed method relies on a crucial hyperparameter, d, which determines both the efficiency and the soundness of the algorithm. How is this parameter selected? Is it possible to estimate it from data?

The authors claim that the proposed method is suitable for learning large networks. However, since there are no theoretical results comparing the number of conditional independence (CI) tests performed by this algorithm to those performed by other relevant algorithms, this claim is not well supported. Additionally, the empirical evidence is too limited to demonstrate the superiority of the proposed algorithm; it is only compared against the PC algorithm. How does it perform against other constraint- and score-based algorithms?

**Questions:**

Please see the above comments.

---

### Meta-Review · Area_Chair_yGKR · 2024-12-23

**Metareview:**

The authors present an exact distributed algorithm for structure learning by divide and conquer. The variables are divided into overlapping subsets called covers such that edges can be identified by learning a graph on each subset separately and then joined in the end. Several reviewers raised concerns, with one expert reviewer flagging significant issues with the evaluation as well as concerns over correctness. Without a response from the authors, this paper needs to be reviewed after a significant revision.

**Additional Comments On Reviewer Discussion:**

There is no response from authors.

---

### Decision · Program_Chairs · 2025-01-22

Reject